

# A framework for generating recommendations based on trust in an informal e-learning environment

Amjad Rehman[1], Adeel Ahmed[2], Tahani Jaser Alahmadi[3], Abeer Rashad Mirdad[1], Bayan Al Ghofaily[1] and Khalid Saleem[2]

[1] Artificial Intelligence & Data Analytics Lab (AIDA) CCIS Prince Sultan University, Riyadh, Saudi Arabia
[2] Department of Computer Science, Quaid-i-Azam University, Islamabad, Pakistan
[3] Department of Information Systems, College of Computer and Information Sciences, Princess Nourah bint Abdulrahman University, Riyadh, Saudi Arabia

Corresponding author
Tahani Jaser Alahmadi,
tjalahmadi@pnu.edu.sa

## ABSTRACT

Rapid advancement in information technology promotes the growth of new online learning communities in an e-learning environment that overloads information and data sharing. When a new learner asks a question, how a system recommends the answer is the problem of the learner's cold start. In this article, our contributions are: (i) We proposed a Trust-aware Deep Neural Recommendation (TDNR) framework that addresses learner cold-start issues in informal e-learning by modeling complex nonlinear relationships. (ii) We utilized latent Dirichlet allocation for tag modeling, assigning tag categories to newly posted questions and ranking experts related to specific tags for active questioners based on hub and authority scores. (iii) We enhanced recommendation accuracy in the TDNR model by introducing a degree of trust between questioners and responders. (iv) We incorporated the questioner-responder relational graph, derived from structural preference information, into our proposed model. We evaluated the proposed model on the Stack Overflow dataset using mean absolute precision (MAP), root mean squared error (RMSE), and F-measure metrics. Our significant findings are that TDNR is a hybrid approach that provides more accurate recommendations compared to rating-based and social-trust-based approaches, the proposed model can facilitate the formation of informal e-learning communities, and experiments show that TDNR outperforms the competing methods by an improved margin. The model's robustness, demonstrated by superior MAE, RMSE, and F-measure metrics, makes it a reliable solution for addressing information overload and user sparsity in Stack Overflow. By accurately modeling complex relationships and incorporating trust degrees, TDNR provides more relevant and personalized recommendations, even in cold-start scenarios. This enhances user experience by facilitating the formation of supportive learning communities and ensuring new learners receive accurate recommendations.

## INTRODUCTION

With the advancement in Internet technologies, many e-learning websites are now deployed, *e.g.*, Khan Academy (www.khanacademy.org/), Udemy (www.udemy.com/), Coursera (www.coursera.org/), and Stack Overflow (www.stackoverflow.com/). These websites offer numerous expertise, online courses, and other information sharing. Recommender systems can help learners select the most appropriate topic or expert based on their previous history of interests. Learners use formal and informal e-learning prospects to create continuous development, knowledge, and skills needed for their career and personal development (*Cao et al., 2021*). Formal e-learning refers to structured and systematic learning settings where institutions organize educational activities and follow a specific curriculum. For instance, online courses offered by universities or platforms like Coursera, where learners aim to achieve official certifications or degrees. Informal e-learning refers to more casual, self-directed learning activities where individuals engage in educational pursuits without the goal of obtaining formal qualifications, such as Stack Overflow, Stack Exchange (*Xu et al., 2024*). An e-learning community consists of friends sharing information in the same domain of interest (*Xu et al., 2024*). Since the knowledge is distributed across the network, e-learning communities focus on building knowledge through collaborative efforts. Therefore, learners need to organize and improve their e-learning abilities to achieve their learning goals (*Huang et al., 2023*; *Xu et al., 2023*). Learners in the digital age have access to a vast number of e-learning resources from a variety of formal and informal settings. Potential learners require some practical strategies to assist them in identifying resources from a diversity of varieties. Learners often retrieve a large amount of knowledge, comments, and e-learning resources, but they cannot filter out the valuable information, resulting in the issue of information overload (*Dang et al., 2024*; *Shahrzadi et al., 2024*). Educational recommender systems have recently attracted much academic attention (*Maqbool et al., 2024*). To meet the goals and needs of the learners, the recommender systems related to education can provide personalized guidance to the learners. *Liu et al. (2021)* presented a novel approach to improving industrial recommender systems by integrating deep matrix factorization with review feature learning. The proposed EDMF model enhances traditional matrix factorization techniques by leveraging review text data to capture more nuanced user preferences and item characteristics (*Liu et al., 2021*). *Liu et al. (2022)* defined a unified graph-based framework integrating multiple perspectives, including user-item interactions, social relationships, and user-generated content. *Li et al. (2021)* designed CARM: a confidence-aware recommender model that accounts for the varying reliability of reviews and user ratings. Recommender systems provide a user with suggested actions, items, or decisions. They can be viewed as tools that process information to extract relevant or significant aspects, thus facilitating the problem of information overload (*Dang et al., 2024*; *Shahrzadi et al., 2024*). The availability of massive amounts of information over the Internet and the number of website visitors pose other main challenges for recommendation systems. Some of these are the cold-start problem (*Ban et al., 2024*; *Yin et al., 2024*; *Parvin, Moradi & Esmaeili, 2019*; *Ahmed et al., 2020*; *Ahmed et al., 2021*) and data sparsity (*Jiang et al., 2024*;

*Zhu, 2023*). The predictive ability of collaborative filtering algorithms is limited due to data sparsity. Concerning cold-start users, standard collaborative filtering techniques fail and cannot generate recommendations (*Jing et al., 2014*).

The collaborative-filtering methods predict the ratings and suggest items to users who have been recommended similar ones (*Montaner, López & De La Rosa, 2003*). Traditional collaborative filtering approaches often lack sufficient historical user behavior data, which limits their ability to generate high-quality recommendations. Additionally, these recommender systems may suffer from changes in trust relationships among users over time (*Deng, Huang & Xu, 2014*). Moreover, the ratings provided by users through these systems may become outdated or noisy (*Ahmed et al., 2021*). Matrix factorization techniques, which rely on factorized latent models (*Xiao et al., 2017*), are used to calculate the accuracy and popularity of preferences (*Koren, 2008*). However, these models have limitations, such as (i) assuming that valued item features are suitable for all users (*Cremonesi, Tripodi & Turrin, 2011*), (ii) failing to capture conditional preferences, and (iii) ignoring social effects on user preferences, including various types of social relationships, peer influence, and the homophily effect (*Lewis, Gonzalez & Kaufman, 2012*).

Data sparsity poses a challenge for collaborative filtering systems, making it hard to produce accurate predictions. Recommender systems, which often operate within specific domains, struggle with sparsity issues and challenges related to new users and items. In situations of data sparsity, users may rate only a few items or may not provide ratings at all (*Khusro, Ali & Ullah, 2016*).

The cold start problem is currently a significant challenge for recommender systems. This issue arises when the system cannot generate a recommended list of items for users, typically due to a lack of sufficient data. The cold start problem is particularly associated with scenarios involving new users, items, or communities (*Khusro, Ali & Ullah, 2016*).

Many types of research have been conducted in this direction, including hybrid models. These models use auxiliary and side information to overcome the cold start problem. As knowledge is disseminated across different networks with various digital formats, knowledge and learning are called "rest in diversity of opinions" (*Sobaih, Palla & Baquee, 2022*). In informal e-learning environments, learners may face the issue of knowledge overload and metacognition. From a learning point of view, connections among e-learning communities play a more important role than those formed through formal education.

Informal e-learning communities are typically located closer to geographic locations (*Neubauer et al., 2011*). E-learning communities can now be developed and maintained online because of the advancement in information technologies. However, organizing information and addressing the problem of information overload (*Shahrzadi et al., 2024*) requires additional time and effort from learners (*Neubauer et al., 2011*). Recommender systems have found great success in social media, giving rise to a novel area of research known as trust-aware recommender systems. In contrast to traditional recommender systems that primarily rely on user-item ratings, trust-aware recommender systems go a

step further by incorporating trust-related information among users alongside rating data (*Ahmed et al., 2022*; *Li et al., 2021*).

The main properties of the proposed TDNR model are:

- The proposed model automatically learns features using deep neural networks to tackle the important rating prediction task. However, the model's performance may suffer if raw features introduce noise. As a result, deep neural networks have become a preferred option for feature representation (*Elkahky, Song & He, 2015*).
- Similarly to the latent factor model (*Koren, 2008*), proposed model leverages user preferences; however, unlike the latent factor model, it also incorporates trust networks and their influences. The proposed model integrates trust information implicitly, enhancing its effectiveness.
- We also aspire to address the issue of developing efficient recommendation models with deep neural architectures to tackle the learner cold start problem. In contradiction to linear models such as matrix factorization (*Koren, 2008*) that only capture the linear relationship in data, the deep neural networks model captures the non-linearity of user-item interactions with nonlinear activation functions such as sigmoid, relu, tanh, *etc.* (*Elkahky, Song & He, 2015*).

## Main contributions

In this article, our contributions are

- We proposed a Trust-aware Deep Neural Recommendation framework called TDNR. It models complex nonlinear relationships for solving learner cold-start problems in an informal e-learning environment and generates recommendations based on experts' trust and their answers.
- We performed tag modelling using latent Dirichlet allocation and identified the tag category for a newly posted question.
- We introduced the concept of trust degree between the questioner and responder and integrated this trust information into the TDNR model to improve recommendation accuracy.
- We also introduced the questioner-responder relational graph from structural information of preferences related to questions and responders and embedded this in the proposed model.

The rest of the article is structured as follows. "Related Work" describes the related work, "Materials and Methods" discusses the proposed approach, "Results" explains the experimental setup and results, and "Conclusions and Discussion" discusses the conclusion.

# RELATED WORK

We categorized the related work into hybrid models used for recommender systems in e-learning, deep learning approaches, and trust-aware recommender systems based on e-learning.

## Hybrid recommendation models for e-learning

Previous research has shown that combining different recommendation approaches, such as collaborative filtering and content-based, into a hybrid approach can be more effective. Both approaches have limitations. Collaborative filtering cannot give better results against new items, but their combination produces better results and solves the common problems of data sparsity and cold start in recommender systems (*Yin et al., 2024*; *Maqbool et al., 2024*). A commercial example of a hybrid recommendation system is Netflix. Another commercial hybrid recommender system is the Google News recommender system (*Peng, Zhao & Hu, 2023*). Some of the hybridization techniques are weighted (*Luo et al., 2022*), cascade (*Imran et al., 2024*), and mixed (*Hussain et al., 2024*). Data sparsity is a crucial challenge for collaborative filtering and content-based recommenders (*Shahzad et al., 2024*), even though collaborative filtering (*Shahzad et al., 2023*) is the most common educational recommendation algorithm. Nowadays, a hybrid approach is getting popular and is not sensitive to these issues (*Sobhanam & Mariappan, 2013*). E-learning recommender systems adopted the best features of a hybrid approach and overcame the drawbacks of existing approaches (*Çano & Morisio, 2017*). In 2021, *Souabi et al. (2021)* surveyed important social and e-learning approaches that enhance learning. *Buder & Schwind (2012)* emphasized how non-technical aspects should be considered when developing personalized educational recommender systems. *Drachsler, Hummel & Koper (2009)* discussed that the recommender systems in informal e-learning differ from formal e-learning. Therefore, the researchers suggested that more efforts are required to produce quality among informal e-learning communities. Learning occurs through conversions, which link collaborative learning's interactive and cognitive elements. Therefore, several studies followed a hybrid approach based on trust and collaborative filtering algorithms because trust is a relationship that establishes and cultivates interactions among users of e-learning communities. Trust is a cognitive and social mechanism that helps people manage various levels of uncertainty and risk (*Bailey, Almusharraf & Almusharraf, 2022*). Due to users' social interactions, learners with similar e-learning preferences may assign varying scores to a post. In an e-learning environment, communities are essential, and the recommender system should consider learners' social connections and affiliations (*Serrano-Iglesias et al., 2019*). In *Ahmed et al. (2021)*, a deep neural network-based custom recommendation model for educational resources is proposed. The model, constructed after a multilayer perceptron-based prediction, demonstrated a consistent reduction in the average absolute error as the number of iterations increased, reaching an average of 0.704, and the loss value stabilized around 0.6. In *Tzeng et al. (2023)*, authors proposed a recommendation system to be integrated into massive open online courses (MOOCs), following a hybrid architecture, where e-learning resources are described by a set of terms

extracted directly from the supporting texts in the MOOC. *Ahmed et al. (2022)* developed a recommendation model for aliexpress based on autoencoder.

## Deep learning-based recommender systems for e-learning

Deep learning has recently succeeded in collaborative filtering-based recommender systems, making it a better choice for e-learning recommender systems (*Zhang et al., 2019*). To assess the learner's ability, *Ahmed et al. (2021)* defined a framework based on a neural network that is integrated with nearest neighborhood collaborative filtering approaches (*Wang et al., 2017*). Several recently proposed deep learning-based recommendation architectures to solve the two critical problems of data sparsity and cold start (*Wei et al., 2017*; *Wang, Wang & Yeung, 2015*). *Shen et al. (2019)* developed a novel approach that combines deep variational matrix factorization with knowledge embedding to enhance recommendation accuracy. The proposed method captures complex user-item interactions and integrates external knowledge, improving prediction performance. In 2019, *Yi et al. (2019)* introduced a technique that combines matrix factorization with implicit feedback embedding to enhance recommendation accuracy. While the approach effectively captures user preferences and improves prediction accuracy, it may still struggle with scalability and computational efficiency, especially in large-scale datasets. Additionally, the reliance on implicit feedback could limit the model's ability to fully understand nuanced user behaviors, potentially impacting recommendation quality in complex scenarios (*Yi et al., 2019*). *Shu et al. (2018)* focused on recommending educational content by analyzing and matching resource features to user preferences. While the approach effectively personalizes learning experiences, it may be limited by its reliance on content descriptions, potentially missing out on capturing the broader context of user needs.

## Trust-aware recommender systems for e-learning

In 2021, *Mohamadrezaei & Ravanmehr (2021)* proposed a trust-based e-learning recommender system that addresses conventional e-learning systems' limitations by incorporating learners' previous interactions and interests. The system clusters users and predicts suitable learning courses using fuzzy clustering and weighted association rules, improving recommendation accuracy and efficiency. The approach, tested on the Moodle dataset, showed reduced MAE and RMSE, leading to more accurate and personalized course recommendations. Trust relationships and fuzzy clustering help to mitigate data sparsity and enhance the recommendations' relevance in large-scale e-learning environments (*Mohamadrezaei & Ravanmehr, 2021*). In *Chen et al. (2017)*, presented an App recommender system for GooglePlay with a deep neural network model. Studies in *Lü et al. (2012)*, *Weerathunga et al. (2021)*, *Dwivedi & Bharadwaj (2013)* developed education recommender systems by integrating trust with collaborative filtering techniques. However, these approaches cannot alleviate the issue of data sparsity in recommender systems. According to *Goodfellow, Bengio & Courville (2016)*, most feedforward deep networks are built on "core parametric function approximation". They observed that supervised learning, such as KNN, logistic regression, and Bayesian classifiers, gave the

best results statistically and computationally compared to unsupervised learning. Most machine learning algorithms in supervised learning overcome the high dimensionality of random variables. However, unsupervised learning algorithms are still not mature (*Goodfellow, Bengio & Courville, 2016*). Feedforward deep network consists of modern neural networks such as recurrent neural networks, convolutional neural networks, and others. These networks are widely used for developing recommendation frameworks (*You et al., 2019*). To enable a hierarchical analysis of user preferences, authors integrated RNN with a unique Temporal Neural Network to capture the historical impact on users' decisions (*You et al., 2019*). In 2019, *Sarwar et al. (2019)* proposed a framework for e-learning based on course summary information that provides personalized recommendations to learners. They integrated MLPs with ontology, which effectively model features. *Shen et al. (2016)* suggested a customized CNN-based algorithm for recommending course resources. This approach extracted the features of learning and student preferences using CNN and predicted the learner ratings. *Mrhar & Abik (2019)* proposed an architecture that facilitates communication between MOOCs and formal e-learning platforms. It allows the system to suggest MOOC courses to learners similar to those they have already taken on the formal platform (*Mrhar & Abik, 2019*). In 2018, *Zhou et al. (2018)* proposed a recommendation framework that found the similarity among the learners, predicted the users' learning path based on LSTM, and recommended the course. This approach reduced the model training time for larger datasets (*Zhou et al., 2018*). *Xu & Zhou (2020)* designed a course recommendation model based on multimodal features using LSTM. *Li et al. (2021)* introduced a novel recommendation model named AutoLFA. This model is a hybrid approach that integrates Autoencoder and Latent Feature Analysis techniques. AutoLFA independently utilizes an autoencoder and an LFA model to create two recommendation models. A limitation of this model that is not addressed by *Guo et al. (2023)* is how varying hyperparameters in the dataset's characteristics might affect the performance of the AutoLFA model.

## Recommender systems based on e-learning

*Khanal et al. (2019)* systematically reviewed machine learning-based recommendation systems in e-learning. They categorized recommendation techniques, machine learning algorithms, and application areas, emphasizing the importance of appropriate validation and evaluation approaches (*Khanal et al., 2019*). *Srivastav & Kant (2019)* conducted a comparative study of deep learning-based e-learning recommender systems, exploring how these techniques addressed challenges like cold-start and sparsity problems. In 2019, *Aeiad & Meziane (2019)* discussed an adaptable and personalized e-learning system using an ontology-based approach. The system integrates VARK learning styles with background knowledge to improve learning outcomes. The system heavily relies on the quality of ontologies and semantic relations, which can be challenging to maintain and update (*Aeiad & Meziane, 2019*). *Ibrahim et al. (2020)* proposed a fog-based recommendation system to enhance e-learning performance, improving personalization and response times. The reliance on fog computing infrastructure may limit the system's applicability in environments where such infrastructure is unavailable (*Ibrahim et al.,*

*2020*). In 2019, *Bhaskaran & Santhi (2019)* introduced a trust-aware hybrid recommender system integrating K-means clustering, Firefly algorithm, and Apriori, showing improved accuracy and efficiency. A limitation is that the model's complexity could be a barrier to implementation in large-scale systems, requiring significant computational resources (*Bhaskaran & Santhi, 2019*). *Wan & Niu (2019)* proposed a self-organization-based recommendation approach, improving adaptability and personalization in e-learning. The study lacks explicit performance evaluations, making it difficult to assess the practical effectiveness of the approach (*Wan & Niu, 2019*). *Kolekar, Pai & Motta (2019)* developed a rule-based adaptive user interface for e-learning systems that adjusts dynamically based on FSLSM learning styles, enhancing user engagement. One of the limitations of this model is that the rule-based system may not scale well with increasing complexity in learning content and user diversity (*Kolekar, Pai & Motta, 2019*).

Based on the challenges mentioned in a literature review, we adopt a hybrid approach in which we integrate the trust and structural information of the learners with deep neural networks to solve the problem of learners cold starting in an informal e-learning environment.

## MATERIALS AND METHODS

### Rationale for proposed framework

The TDNR framework addresses challenges in e-learning environments, particularly the cold-start problem where new users (learners) have insufficient interaction history for generating accurate recommendations. Traditional recommendation systems struggle with this issue due to data sparsity and the inability to effectively model complex, nonlinear relationships between users and items. By integrating trust and structural information of learners into a deep neural network, TDNR aims to provide more accurate and personalized recommendations. The rationale for this approach lies in the observation that trust relationships and historical interactions between learners and experts can significantly enhance recommendation accuracy. As a social and cognitive mechanism, trust helps manage uncertainties and risks associated with recommendations, thereby improving the system's effectiveness.

Deep neural networks (DNNs) and trust-sensitive systems are particularly effective in addressing the cold start problem in recommendation systems. DNNs excel at capturing complex, nonlinear relationships in data, allowing them to generalize from existing patterns even when there is limited interaction data for new users or items. This capability is crucial in cold start scenarios, where direct data is sparse. Trust-sensitive systems, on the other hand, leverage social trust relationships to make recommendations. When new users have little to no interaction history, their trust connections with more established users can inform the system of likely preferences. Combining DNNs' pattern recognition with trust-sensitive insights enables more accurate and personalized recommendations for users in cold start situations.

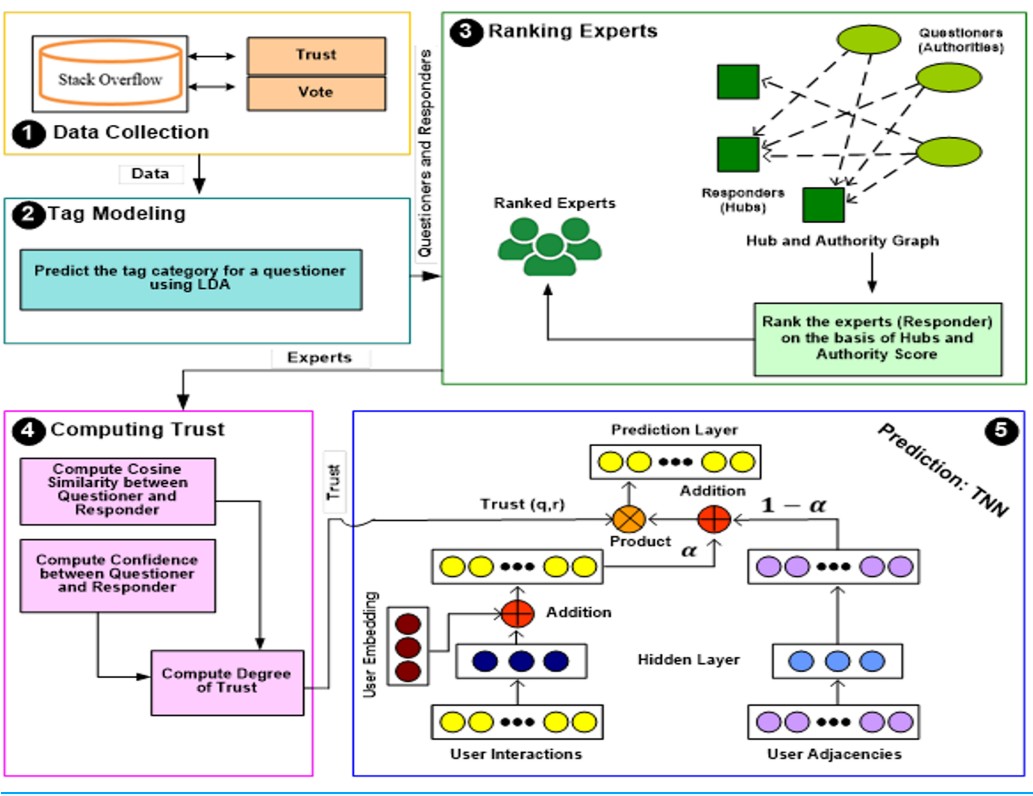

**Figure 1 Proposed architecture of TDNR.**

## Proposed framework

We propose a model called Trust-aware Deep Neural Recommender in which we combine the learner preferences information with neural networks (*He et al., 2017*) and generate recommendations based on trust present among the responders (experts). The architecture of the TDNR model is shown in Fig. 1. This model consists of four modules. The first module, Tag Modelling, assigns a particular tag to a new question using the LDA approach. The second module, Ranking Experts, filters out experienced responders based on hub and authority scores. The third module is Computing Trust, which estimates the trust intensity among responders, and the fourth module predicts the voting for an active questioner.

### Problem definition

To solve a problem of a user (questioner) cold start in an e-learning environment, we formulate the problem as, "If a new user (questioner) comes and asks/posts a question in an e-learning environment, then how will a system recommend the expert based on tagging information and the vote?" For each questioner q ∈ Ne, the goal is to predict the vote for an answer related to Ae on an e-learning community by incorporating the structural information adopted by trusted neighbors of q.

Let us assume that $Q = \{q_1, q_2, \ldots, q_n\}$ be the set of all questioners, $A = \{a_1, a_2, \ldots, a_m\}$ be the set of all answers provided by experts, $T = \{t_1, t_2, \ldots, t_k\}$ be the set of tags associated with questions and $T = \{t_1, t_2, \ldots, t_k\}$ be the set of votes given to answers.
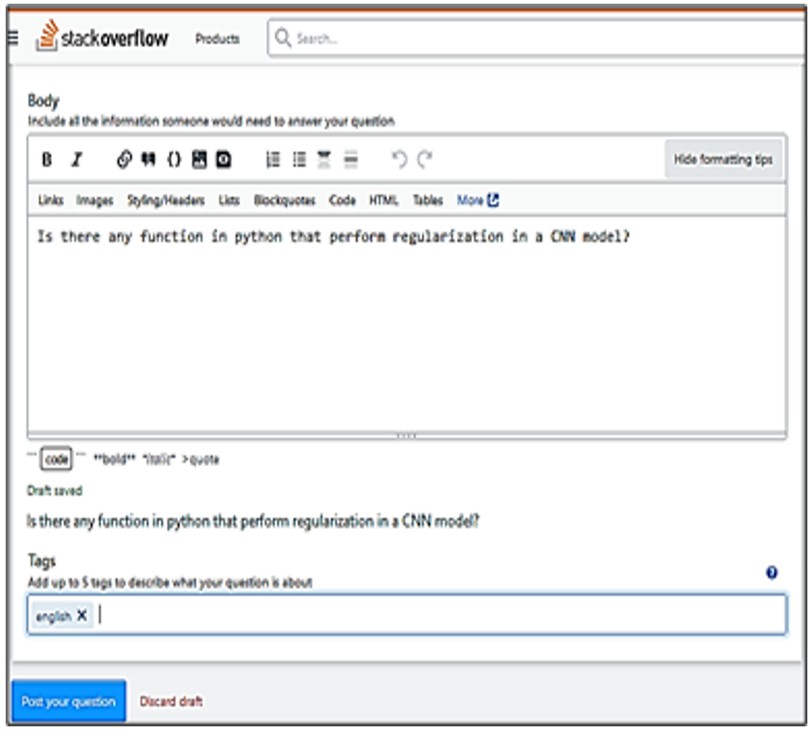

**Figure 2 Stack overflow interface.**

Suppose a new user $q_{new}$ posts a question $q$ in the e-learning environment. The question $q$ is associated with a tag $t_i \in T$ based on its content. No historical data is available for $q_{new}$, making it a cold start problem.

Let $Ne(q_{new})$ represent the set of trusted neighbors of $q_{new}$ (*i.e.*, other users who have interacted with similar questions or have a trust relationship with $q_{new}$). The goal is to predict the vote $v_q^a$ for the answer $a_j$ based on the tag $t_i$ and the votes given by the trusted neighbors in $Ne(q_{new})$ for a cold start user $q_{new}$.

### Tag modeling

The motivation behind tag modeling is that in an e-learning environment such as Stack Overflow, if a new questioner enters the wrong tags at the time of posting a question, then the system posts a question under the wrong tag category, and learner (questioner) may get suffer from wrong answers from experts. Figure 2 shows the screenshot taken from the web interface of Stack Overflow for the new learner.

Here, the new learner posts the question, "*Is there any function in Python that performs regularization in a CNN model*?" and enters the wrong tag, '*English*' instead of Python or deep learning, so the system will post this question under the wrong tag categories. Due to this manual false selection, the performance of the system is affected. We predicted the tag category for a question that an active questioner posts and categorized it according to given tag categories using latent Dirichlet allocation (LDA). We also observed how each question belongs to each tag category based on the text in a question. Figure 3 shows some possible tags present in a Stack Overflow dataset.

**Figure 3  List of tags available at stack overflow dataset.**

We use LDA (*Ali et al., 2022*) to solve the problem of assigning a tag to the question. The questions are grouped according to the tags. We treat each question as a document. LDA infers the structure of hidden topics using the observed words. There are Q documents (questions) in an e-learning *corpus* C denoted by $C = \{q_1, q_2, \ldots .q_Q\}$ and each question $q$ has N words denoted by $w = \{w_1, w_2, \ldots .w_N\}$. All words are categorized into k tags (topics) using the approach discussed in *Ali et al. (2022)*. The key assumption is that each document represents a question on Stack Overflow and is a mixture of topics, and each word in the document is drawn from one of these topics.

LDA assumes that for each document $q$ (a question in our case), there is a distribution over topics denoted by $\theta_q$, which is drawn from a Dirichlet distribution with parameter $\alpha$:

$$\theta_q \sim \text{Dirichlet}(\alpha)$$

Similarly, for each topic (tag) $k$, there is a distribution over words denoted by $\beta_k$, which is also drawn from a Dirichlet distribution but with parameter $\eta$:

$$\beta_k \sim \text{Dirichlet}(\eta)$$

Here, $\alpha$ and $\eta$ are hyperparameters that control the sparsity of the distributions $\theta_q$ and $\beta_k$ respectively.

For each word $w$ in a document (question) $q$, a topic $z$ is chosen from the document's topic distribution $\theta_q$ is represented as

$$z_{qn} \sim \text{Multinomial}(\theta_q)v$$

where $z_{qn}$ indicates the topic assigned to the nth word in the document $q$.

Once a topic $z_{qn}$ is chosen, the actual word $w_{qn}$ is drawn from the corresponding topic's word distribution $\beta_{z_{qn}}$:

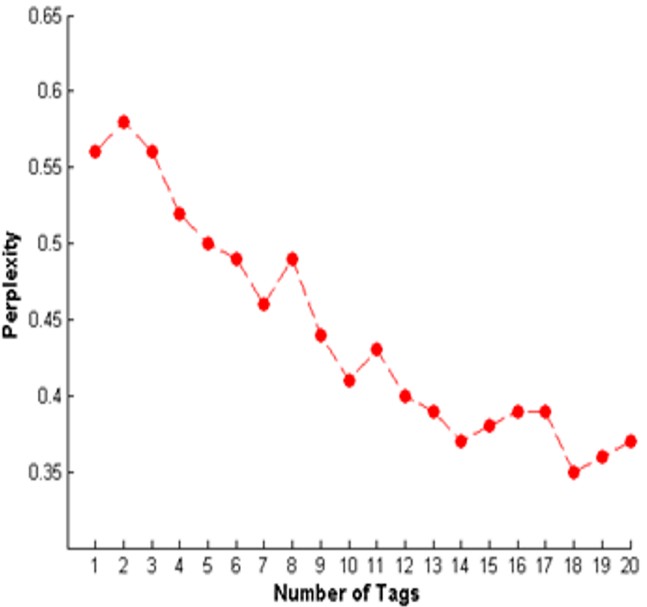

**Figure 4 Evaluation of tag modeling using perplexity score.**

$$w_{qn} \sim \text{Multinomial}\left(\beta_{z_{qn}}\right).$$

This step generates the observed word from the selected topic's word distribution.

We used the perplexity metric to evaluate tag modeling. Perplexity tells us how well the model predicts the samples. Figure 4 shows the evaluation of the tag model based on the perplexity score depending on the number of tags. It is evident from Fig. 4 that the lowest perplexity value is for 18 tags, so we trained the LDA model with 18 tags (topics).

### Ranking experts

The output of the tag modeling phase is identifying a tag for a particular question about an active questioner. After identification of the tag, we filter out all the questioners and responders related to that tag and then construct a hub authority graph among them. HITS algorithm performs ranking and filters the most experienced experts and their answers (votes) (*Easley & Kleinberg, 2010*). For computing a hub and authority score, we used the formulas proposed by *Easley & Kleinberg (2010)*, based on a bipartite graph that consists of questioners as authorities and responders as hubs, as shown in Fig. 5. In a given graph, we assume that node '1343' is an authority (questioner) node and other nodes are called hubs (responders). We chose those responders as experts for an active questioner with a hub score greater than or equal to the threshold value of 0.5. Therefore, the experts for an active questioner '1343' in the '3D Graphics Model' tags are '1131', '2666', '57625', and '1527'. In this way, we ranked the experts based on the hub score.

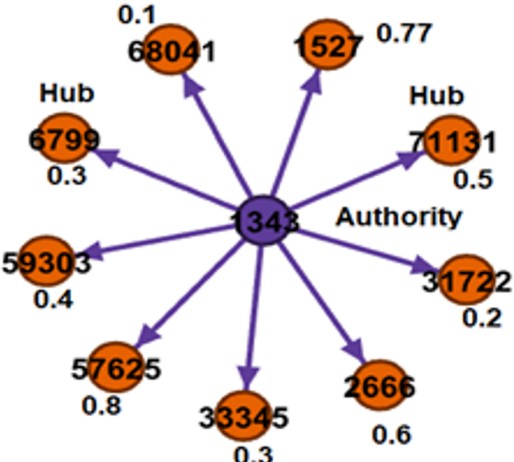

**Figure 5  Hub and authority graph for '3D Graphics Model' Tag: A node in blue color represents authority, and nodes in orange color represent a hub.**

### Computing trust

After ranking the experts, we defined the trust degree $T(q, r)$ between questioner q and responder (expert) r of the e-learning community by combining similarity and confidence using Eq. (1).

$$T(q, r) = \frac{2 \times similarity(q, r) \times confidence(r|q)}{similarity(q, r) + confidence(r|q)}. \tag{1}$$

Equation (1) describes that, $T(q, r)$ may not be the same as $T(q, r)$ that is true in real-life scenarios (*Bedi & Sharma, 2012*). Here, similarity(q, r) is the Pearson correlation coefficient for computing similarity between the questioner q and responder r, and this similarity can be computed using Eq. (2).

$$similarity(q, r) = \frac{\sum \left(s_{q,a} - \bar{s}_q\right)\left(s_{r,a} - \bar{s}_r\right)}{\sigma_q \sigma_r} \tag{2}$$

where $s_{q,a}$ and $s_{r,a}$ denote the vote of questioner q and responder r for answer 'a', respectively. $\bar{s}_q$ is the average voting of questioners q and $\bar{s}_r$ shows the average voting of responder r, respectively. $\sigma_q$ and $\sigma_r$ are the standard deviations of voting given by q and r, respectively. The similarity value lies between [0, 1]. If $similarity(q, r) \leq 0$, shows that the q and r are correlated.

The confidence(r|q) is the confidence that represents how much a questioner q is interested in responder r and vice versa. The confidence can be computed using Eq. (3).

$$confidence(r|q) = \frac{No.\ of\ answers\ voted\ by\ q\ and\ r\ in\ common}{No.\ of\ answers\ voted\ by\ q}. \tag{3}$$

The confidence(r|q) is large if the voting overlap between q and r is high; otherwise, confidence(r|q) has less value.

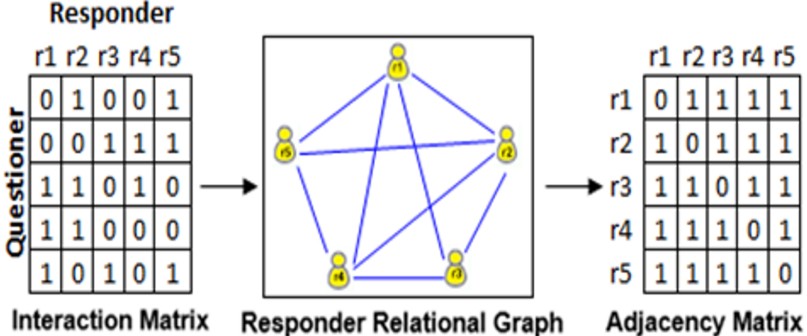

**Figure 6 Constructing responder relationship graph from questioner behaviours.**

In TDNR, we used trust as an outcome for learner interaction and a basis for the recommendation model. Trust information significantly influences the recommendation process in the TDNR model. The model incorporates trust degrees into the final prediction of how a questioner would vote on an answer provided by a responder. This integration of trust helps the model better predict the questioner's preferences, particularly in cold-start scenarios where traditional recommendation methods might struggle due to a lack of historical data. By leveraging trust, the TDNR model can provide more personalized and accurate recommendations, improving the overall effectiveness of the e-learning platform. This approach ensures that even with minimal direct interaction data, the system can still make informed recommendations by relying on the trust relationships within the community, thereby addressing the challenges associated with the cold-start problem.

### Prediction

Figure 6 illustrates the responder relational graph construction process. The graph is formed based on the interaction matrix, linking two responders when they share the same row in the interaction matrix, effectively representing the accurate coexistence of responders. For instance, if the first questioner engages with both responder r2 and responder r5, an edge exists between node r2 and node r5. Easily derived from the responder graph, the adjacency matrix indicates whether pairs of responders are adjacent or not. Let $A \in R^{N \times N}$ be the adjacency matrix of the responder graph. Let $A_{(i)} = \{A_{i1}, A_{i2}, \ldots .. A_{iM}\}$ denote the ith row, indicating the relationships between responder r and remaining responders. The adjacency matrix keeps the structural information of the responder relational graph. The Trust-aware Neural Network (TNN) prediction module consists of two parallel neural networks, the outputs of which are weighted and summed up for final prediction. A TNN learns responder representations from both questioner-responder and responder-responder structural Information. The final scoring layer is formed by combining the predictions from both networks.

#### 1) Input and embedding layer

The input of TNN consists of responder interaction behavior, denoted as $B_{(i)}$ and responder structural information denoted as $A_{(i)}$. There are two main reasons for using $B_{(i)}$ and $A_{(i)}$ as inputs: first, the responder index alignment is maintained, allowing for the

incorporation of responder graph structure. As a result, the two sub-networks can be trained simultaneously. Secondly, the use of $B_{(i)}$ and $A_{(i)}$ is based on the concept of responder similarity. Interaction distributions per responder are typically more stable than per questioner (*Atawneh et al., 2020*). Additionally, there is a responder embedding, $P_i \in \mathbb{R}^d$, illustrated in Fig. 1. $P_i \in \mathbb{R}^{N \times d}$ represents a learnable parameter and can be considered a responder latent factor in matrix factorization.

### 2) Hidden layer

Based on the provided minimal inputs $B_{(i)}$ and $A_{(i)}$, we apply non-linear transformations to convert them into compact, high-density representations. These transformed states are represented as $H_{(i)}^B$ for $B_{(i)}$ and $H_{(i)}^A$ for $A_{(i)}$. The methodology for this transformation is modeled in Eqs. (4) and (5).

$$H_{(i)}^B = f\left(B_{(i)} W_1^B + b_1^B\right) \tag{4}$$

$$H_{(i)}^A = f\left(A_{(i)} W_1^A + b_1^A\right) \tag{5}$$

where $W_1^B \in \mathbb{R}^{M \times d}$, $W_1^A \in \mathbb{R}^{N \times d}$, $b_1^B \in \mathbb{R}^d$, $b_1^A \in \mathbb{R}^d$ are weights and biases. f is the activation function illustrating the advantageous impact of introducing non-linearity on the model's efficacy. These two dense embeddings encode responder information from different perspectives, with $H_{(i)}^B$ for responder relationships.

### 3) Prediction layer

In the TNN framework, every individual sub-network includes a layer for intermediate predictions. The notation $Z_{(i)}^B \in \mathbb{R}^M$ represents the scores derived from the structure of information exchange between the questioner and responder. Similarly, $Z_{(i)}^A \in \mathbb{R}^M$ signifies the scores expected from the interactions and relationships between different responders, as written in Eqs. (6) and (7).

$$Z_{(i)}^B = \sigma\left(H_{(i)}^B W_2^B + b_2^B + P_i\right) \tag{6}$$

$$Z_{(i)}^A = \sigma\left(H_{(i)}^A W_2^A + b_2^A\right) \tag{7}$$

where $W_2^B \in \mathbb{R}^{d \times M}$, $W_2^A \in \mathbb{R}^{d \times M}$, $b_2^B \in \mathbb{R}^M$, $b_2^A \in \mathbb{R}^M$ are weights and biases. $\sigma$ represents sigmoid function. The embedding of the responder, denoted as $P_i$ It is input into Eq. (6), which functions as a unique bias adapted for voting prediction. Specifically, $W_2^B$ and $W_2^A$ can be considered as latent factors associated with the questioner in matrix factorization.

In Eq. (8), we incorporated the trust intensity that is $T(q, r)$ as derived using Eq. (1), as the product of the trust between questioner q and their neighbor r. The ultimate score is derived by merging $Z_{(i)}^B$ and $Z_{(i)}^A$, taking into account a weight factor α along with a measure of trust intensity. Consequently, we have

$$\hat{B}_{(i)} = \alpha \, Z_{(i)}^B + (1-\alpha) \, Z_{(i)}^A \times \prod_{r \in N_q} T_{qr} \tag{8}$$

where $\hat{B}_{(i)} \in \mathbb{R}^M$ represents the preference of all questioners over responders r, including

---

**Algorithm 1** TNN: prediction.

**Input:** Interaction behavior matrix: $B_{(i)}$, Structural information matrix: $A_{(i)}$ for each responder, Trust intensity $T(q, r)$ between questioner $q$ and responder $r$, Weight factor: α

**Output:** Prediction score $B_{(i)}$

1. **for** each responder $i$:
2.     Input $B_{(i)}$ and $A_{(i)}$
3.     Initialize responder embedding $P_i$
4. **end for**
5. **for** each responder $i$:
6.     Transform $B_{(i)}$ to $H_{(i)}^B$ using Eq. (4).
7.     Transform $A_{(i)}$ to $H_{(i)}^A$ using Eq. (5).
8. **end for**
9. **for** each responder $i$:
10.     Compute $Z_{(i)}^B$ using Eq. (6).
11.     Compute $Z_{(i)}^A$ using Eq. (7).
12.     Compute predicted score $\hat{B}_{(i)}$ using Eq. (8)
13.     Compute $P(B|\Theta)$ using Eq. (9).
14.     Compute cross-entropy loss $\ell(\Theta)$ using Eq. (10).
15.     Train the network by minimizing cross-entropy loss $\ell(\Theta)$.
16.     Use regularization techniques like dropout and $\ell_2$ norm and apply an adaptive gradient algorithm for parameter learning.
17. **end for**
18. **return** $\hat{B}_{(i)}$

---

$\hat{B}_{1r}, \hat{B}_{2r}, \ldots \ldots B_{Mr}$. The factor $\alpha \in [0, 1]$ determines the extent to which various structural information influences the prediction outcome.

We have given TNN a probabilistic interpretation. The probability of all feedback in the dataset is represented in Eq. (9).

$$P(B|\Theta) = \prod_{(q,r)\epsilon B^+} \hat{B}_{qr} + \prod_{(q,r)\epsilon B^-} (1 - \hat{B}_{qr}) \times \prod_{r \in N_q} T_{qr} \tag{9}$$

where $B^+$ represents the set of questioner-responder pairs where $B_{qr} = 1$, and $B^-$ represents the set where $B_{qr} = 0$. By applying the negative logarithm to the likelihood, the resulting expression as modeled in Eq. (10) is a cross-entropy loss.

$$\ell(\Theta) = -\sum_{(q,r)\epsilon Y} B_{qr} \log(\hat{B}_{qr}) + (1 - B_{qr}) \log(1 - \hat{B}_{qr}) \tag{10}$$

In Eq. (10), $\Theta$ encompasses the model's parameters, such as the responders' weights, biases, and embeddings. To regulate these parameters in $\Theta$, the $\ell_2$ norm, along with a regularization rate $\lambda$, is employed. The dropout method, which is set to zero for the output of each hidden neuron with a 50% probability, is utilized for regularization. The network undergoes end-to-end training to minimize cross-entropy loss. Specifically, the adaptive gradient algorithm (*Duchi, Hazan & Singer, 2011*) is used for parameter learning, as it helps in reducing the need for extensive tuning of the learning rate.

Algorithm 1 describes the prediction process adopted by Trust-aware Neural Network.

### Time complexity

To analyze the computational complexity of Algorithm 1 line by line, we performed complexity analysis based on the number of responders, denoted as $n$, and the size of the matrices $B_{(i)}$ and $A_{(i)}$ is represented as $m \times m$. Line 1 for loop runs for each responder, so its time complexity is $O(n)$. Line 2: Inputting $B_{(i)}$ and $A_{(i)}$ can be considered $O(1)$. Line 3: Initializing the responder embedding $P_i$ is $O(1)$. The loop runs for each responder, making the total complexity for lines 5–7 $O(n \times m^2)$. Lines 9–16 are executed for each responder $i$, contributing $O(n)$. Each operation within this loop (including forward pass, backward pass, and parameter updates) contributes ($O(\text{cost\_per\_training})$). Since the training process involves multiple iterations k, the overall complexity for these lines becomes ($O(k \times n \times \text{cost\_per\_training})$). Here $k$ represents the total number of iterations or epochs over which the model is trained. Typically, deep learning models require multiple passes over the entire training dataset to converge to a minimum loss. cost\_per\_training represents the computational cost of a single iteration of training, which includes the cost of forward propagation (computing outputs) and backpropagation (updating weights). This cost is influenced by the size of the model (number of parameters, which depends on the dimensions of the input and hidden layers) and the complexity of the operations (such as matrix multiplications, non-linear activations, and regularization). Lines 9–16 typically involve multiple iterations over the entire data, so this could be O(k × n × cost\_per\_training), where k is the number of iterations. Line 18 takes $O(1)$ time. Thus, the overall time complexity of Algorithm 1 is $O(n \times m^2)$.

### Time complexity analysis

Consider implementing the Trust-aware Neural Network (TNN) model from Algorithm 1 for a recommendation system on a large e-learning platform. Suppose the platform has 10,000 responders (users who answer questions), with each responder represented by a feature vector of size 100 and an embedding dimension of 50. The model has trained over 100 iterations (epochs). Each operation, including matrix multiplications and non-linear transformations, has a complexity of $O(m \times d)$, where $m$ is the feature size, and $d$ is the embedding dimension. The total cost per iteration for each responder, combining forward pass, loss computation, and parameter updates, sums up to $O(25,000)$. With 10,000 responders, the complexity for one iteration is $O(250,000,000)$, leading to a total complexity of $O(25 \times 10^9)$ for 100 iterations. This complexity analysis shows that the algorithm incurs a significant computational cost, especially when scaled to real-world

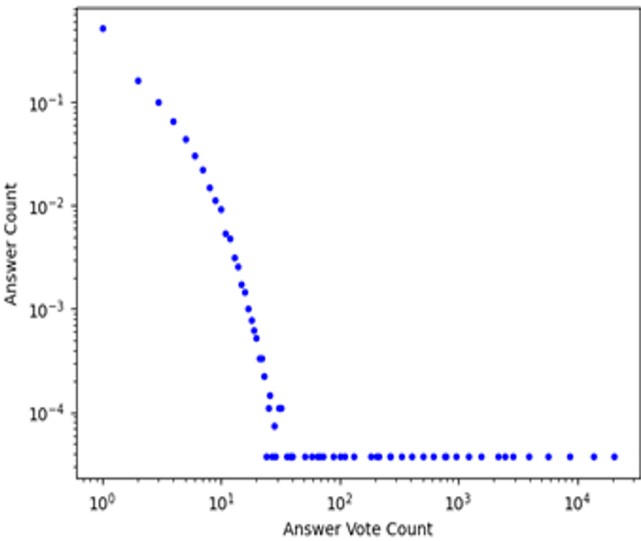

**Figure 7 Number of votes for answers present in a stack overflow dataset.**

scenarios with large datasets. If the platform doubled the number of responders or increased the feature size, the complexity would grow exponentially.

## RESULTS

We scrapped the data from Stack Overflow using Google Chrome API and collected it publicly from 'www.stackoverflow.com', an online e-learning environment. We collected about 236,789 answers and 15,098 unique users. The dataset is sparse because each questioner has given answers to less than fifty questions. The Stack Overflow dataset consists of the following attributes: QuestionerID, QuestionID, QuestionText, Tags, AnswerID, AnswererID, Vote, and AnswerTime. Figure 7 shows the number of votes for answers present in a dataset. The vote for the answer lies within the range of 1 to 10. Within a social network, users indicate their trust towards others using a binary form [0, 1]. A value of one denotes trust, while a lack of trust is represented by 0. In the Stack Overflow dataset, no explicit trust relations are available among questioners and responders, so we generated implicit trust relationships among users. For the sampling of data used in TDNR model, we split the data on a ratio of 30% for test set and 70% for the training set, related to all views of data.

We have performed the experiments on Core i-7 machine with 64 GB RAM related to proposed model and baselines. Our experimental study utilized different machine learning libraries or frameworks such as TensorFlow, pandas, sci-kit learn, and matplotlib.

### Views of data

We evaluated the TDNR model based on different views of data, including:

*All Learners:* A set of learners is assigned to vote for answers from 1 to 10.

*Cold Start Learners:* These learners are assigned to vote for less than three answers. There are about 3.33% cold start learners in the Stack Overflow dataset.

*Heavy Learners:* A set of learners who voted higher than five answers. In this dataset, the rich learners are about 5.55%.

## Evaluation metrics and baselines

TDNR's performance is assessed using the root mean square error (RMSE) and mean absolute error (MAE) metrics (*Ahmed et al., 2020*). The evaluation involves the application of the RMSE metric, calculated in Eq. (11).

$$RMSE = \sqrt{\frac{\sum_{q,a} (v_{qa} - \hat{v}_{qa})^2}{N}} \qquad (11)$$

$v_{qa}$ is the actual vote, $\hat{v}_{qa}$ is predicted to vote for an answer about active questioner $q$. $N$ is the total number of votes under evaluation. The MAE metric is represented in Eq. (12).

$$MAE = \frac{1}{N} \sum_{a=1}^{N} |v_{qa} - \hat{v}_{qa}|. \qquad (12)$$

According to *Deng, Huang & Xu (2014)*, F-measure is defined in Eq. (13).

$$F\_measure = \frac{2 \times precision \times coverage}{precision + coverage}. \qquad (13)$$

We have also evaluated the results of TDNR model by comparing it with rating-based, trust-based, and deep learning-based methods.

**BasicMF:** *Koren (2008)* introduced a method of matrix factorization based on user and item features and predicted the ratings of an item.

**SVD:** *Koren, Bell & Volinsky (2009)* proposed a temporal-based recommender model combining neighborhood model and matrix factorization to improve the prediction accuracy.

**SocialMF:** *Jamali & Ester (2010)* proposed an approach called SocialMF for solving a cold start problem in trust-based recommendations.

**SoReg:** *Ma et al. (2011)* developed a recommendation framework and improved the prediction accuracy.

**NNMF:** *Dziugaite & Roy (2015)* combined the multilayer neural networks with a matrix factorization approach and proposed a new model for rating prediction.

**TrustCTR:** *Ahmed et al. (2020)* proposed a rating-based recommendation framework using implicit and explicit trust relations present among the users of social networks.

**RSTE:** *Ma, King & Lyu (2009)* proposed a hybrid recommendation model called RSTE that is based on social network and matrix factorization.

**LOCABAL:** This technique combines the local and global social media context to produce recommendations (*Tang et al., 2013*).

**Table 1 Parameter settings for TDNR and baselines.**

| Methods | learning rate ($\lambda$) | $\alpha$ | Hidden neurons | Latent dimensions |
|---|---|---|---|---|
| BasicMF | 0.03 | – | – | – |
| SVD | 0.05 | – | – | – |
| SoReg | 0.1 | – | – | – |
| Social MF | 0.05 | 0.3 | – | – |
| RSTE | 0.001 | 0.6 | – | – |
| LOCABAL | 0.03 | 0.4 | – | – |
| TrustCTR | 0.03 | 0.3 | – | 50 |
| NNMF | 0.01 | – | 100 | 10 |
| TDNR | 0.001 | 0.2 | 100 | 100 |

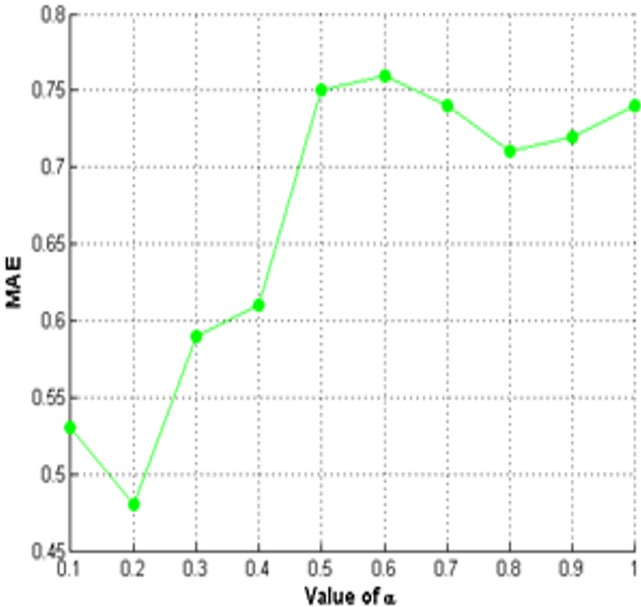

**Figure 8 MAE Results for varying number of $\alpha$ on stack overflow for all learners using TDNR model.**

## Parameter settings and analysis

Table 1 describes the parameters used in our experimental study. These parameters are used for baselines and proposed model. For the TDNR model, these parameters are chosen after performing comprehensive experiments over 100 iterations. The learning rate of TDNR is fixed to 0.001 without further tuning since we adopt the adaptive gradient learning method. Regularization rate is tuned amongst {0.1, 0.2, 0.3, 0.04, 0.09}. Regularization helps penalise larger coefficients, ensuring that the model generalizes well to unseen data. The dimensions of the hidden layer are adjusted within the range of {100, 200, 300, 400, 500}.

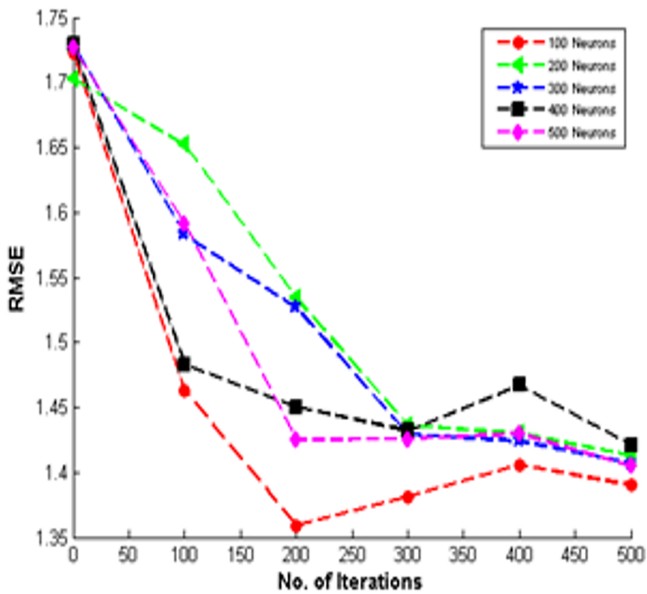

**Figure 9  Results of RMSE with various hidden neurons for all learners on stack overflow.**

## Impact of α on TDNR

We analyze the performance of the TDNR model for all learners, as shown in Fig. 8, based on α. Setting α to 1 means only questioner-responder structural information is used, while 0 means only responder-responder structural information is retained. If the value of α lies between 0 and 1, then it means that both of the structural information play a role. We obtained the best value of MAE with $\alpha = 0.2$, and it determined that the questioner-responder structural information significantly improves vote prediction.

## Impact of hidden neurons on TDNR

We investigate how deep learning models help produce trust-based recommendations in an informal e-learning environment because there is minimal research on generating recommendations *via* deep neural networks. In this way, we explored the neural networks with different hidden dimensions on the Stack Overflow dataset for all learners, as shown in Fig. 9.

The hidden neurons of 100, 200, 300, 400, and 500 are evaluated using the proposed model. This range was explored to determine the best network depth and width that could effectively capture the nonlinear relationships between the questioners and responders. It is observed that the performance of TDNR can be improved by increasing the hidden neurons. In training the model, the neural networks acquire a hidden representation by reducing the loss in the reconstruction phase. Consequently, adding more hidden neurons can enhance the model's effectiveness. Additionally, it has been noted that merely concatenating feature vectors of questioner and responder is insufficient for accurately modeling their interactions. Hence, a transformation using hidden neurons is necessary.

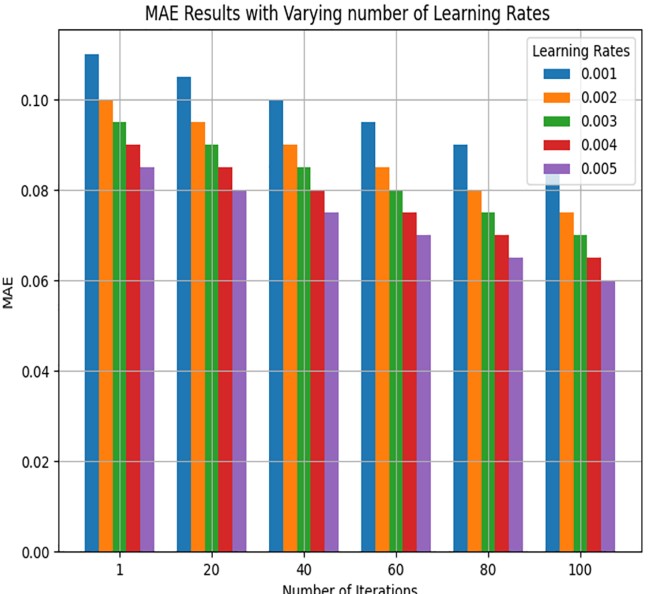

**Figure 10 MAE Results with varying learning rates for all learners on Stack Overflow.**

**Table 2 Comparison of TDNR with baselines.**

| Methods | Error metrics | All learners | Cold start learners | Heavy learners |
|---------|---------------|--------------|---------------------|----------------|
| BasicMF | MAE | 0.802 | 0.741 | 0.816 |
| | RMSE | 1.152 | 0.970 | 0.981 |
| | F-measure | 0.827 | 0.868 | 0.849 |
| SVD | MAE | 0.793 | 0.712 | 0.801 |
| | RMSE | 1.134 | 0.926 | 0.965 |
| | F-measure | 0.841 | 0.877 | 0.836 |
| RSTE | MAE | 0.923 | 0.804 | 0.854 |
| | RMSE | 1.207 | 1.107 | 1.062 |
| | F-measure | 0.813 | 0.834 | 0.801 |
| SoRec | MAE | 0.822 | 0.731 | 0.859 |
| | RMSE | 1.199 | 1.021 | 1.018 |
| | F-measure | 0.828 | 0.852 | 0.850 |
| SocialMF | MAE | 0.847 | 0.723 | 0.825 |
| | RMSE | 1.166 | 1.004 | 1.054 |
| | F-measure | 0.825 | 0.865 | 0.847 |
| LOCABAL | MAE | 0.914 | 0.716 | 0.850 |
| | RMSE | 1.213 | 0.997 | 1.033 |
| | F-measure | 0.802 | 0.868 | 0.831 |
| TrustCTR | MAE | 0.821 | 0.799 | 0.722 |
| | RMSE | 1.100 | 0.917 | 0.928 |
| | F-measure | 0.818 | 0.853 | 0.826 |

| Table 2 (continued) | | | | |
|---|---|---|---|---|
| Methods | Error metrics | All learners | Cold start learners | Heavy learners |
| NNMF | MAE | 0.877 | 0.885 | 0.799 |
| | RMSE | 1.141 | 1.112 | 0.982 |
| | F-measure | 0.874 | 0.845 | 0.825 |
| TDNR | MAE | 0.957 | 0.909 | 0.868 |
| | RMSE | 1.224 | 1.156 | 1.076 |
| | F-measure | 0.892 | 0.888 | 0.867 |
| Improve | | 8.421% | 15.552% | 5.662% |

## Impact of learning rate on TDNR

Figure 10 indicates that a learning rate of 0.001 is ideal, resulting in enhanced performance of the proposed model following 100 iterations. The chosen learning rate of 0.001 was selected after testing various rates, balancing the model's convergence speed and stability. This rate was optimal, particularly when using the adaptive gradient method, which adjusts learning rates during training, thereby reducing the need for extensive manual tuning. The convergence analysis shows that increasing the number of iterations can enhance model accuracy. This evidence supports the effectiveness of cross-entropy log loss in optimizing the proposed objective function.

## Comparison with baselines

We observed that the TDNR model performs very competitively and achieves the best performance across the Stack Overflow dataset. Table 2 shows that our TDNR model gives better results than other state-of-the-art methods regarding MAE, RMSE, and F-measure in the view of All Learners. In the experiments performed for All Learners, the TDNR model converged after 70 out of 100 iterations, whereas other baseline methods converged after 81 iterations. For All Learners and Heavy Learners, the proposed model gives improved results compared to baselines and converges after 75 and 78 iterations for All Learners and Heavy Learners, respectively. The performance of the TDNR model is also evaluated by computing the mean improvement value of MAE concerning different user views. We have found an improvement rate of 8.42%, 15.55%, and 5.66% for All Learners, Cold Start Learners, and Heavy Learners, respectively.

Table 2 reports that the model shows the main improvement of about 15.55% for Cold Start Learners.

## Convergence analysis

We have performed convergence analysis for All Learners and Cold Start Learners, as shown in Figs. 11 and 12, respectively. Figures 11 and 12 show that the TDNR model converged to lower MAE values in later iterations, similar to other methods. It is also observed that the TDNR model can outperform traditional methods such as BasicMF, SVD, SoReg, and SocialMF and achieve better MAE values than neural network-based approaches such as NNMF, confirming the effectiveness of incorporating trust and

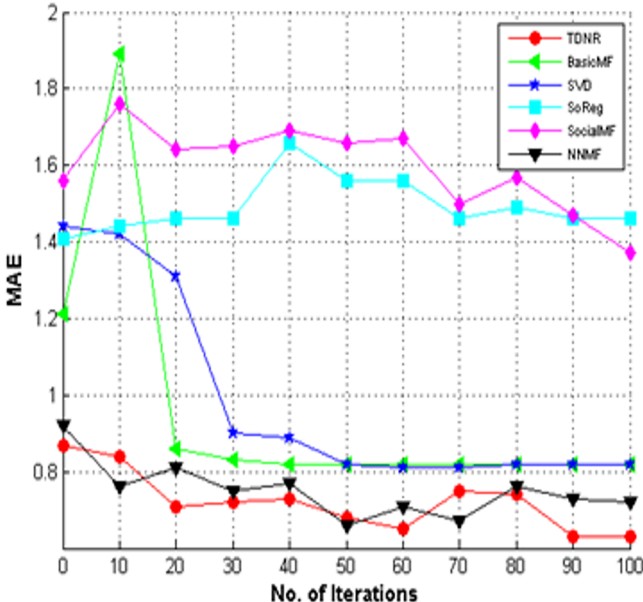

**Figure 11 Convergence analysis of TDNR with baselines for all learners.**

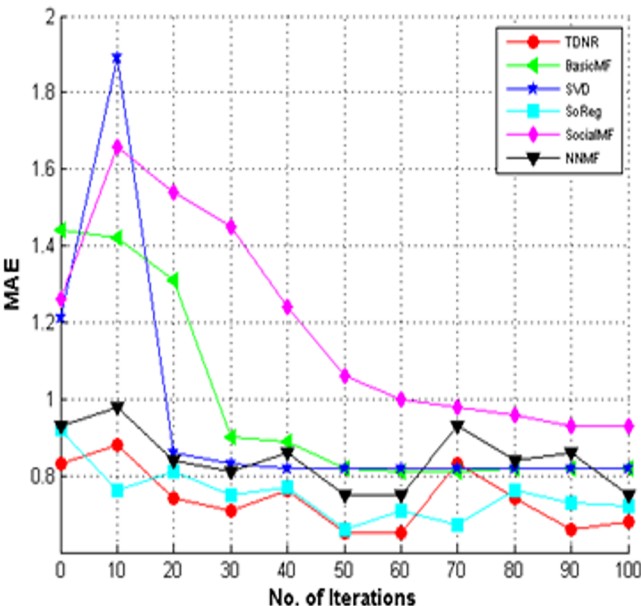

**Figure 12 Convergence analysis of TDNR with baselines for cold start learners.**

structural information. Unlike these baseline approaches, TDNR leverages the non-linearity inherent in neural networks and incorporates two distinct structural elements, enhancing the model's representative capacity.

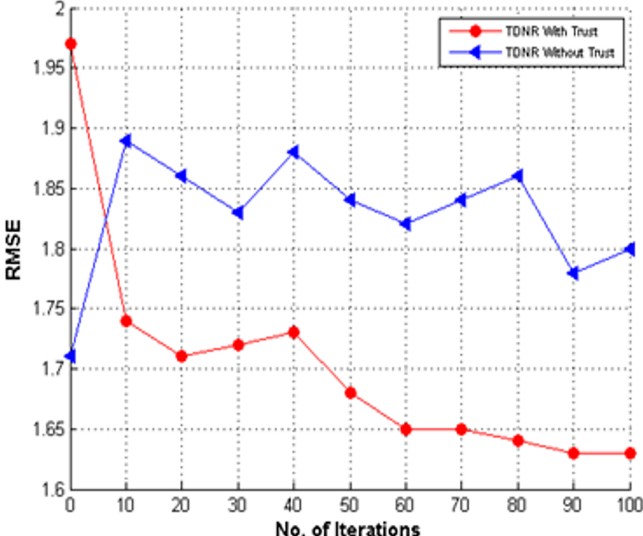

**Figure 13 Performance of TDNR with trust and without trust on stack overflow.**

## Impact of trust on TDNR

We also compared the performance of the proposed TDNR model by incorporating trust and without-trust information. Figure 13 shows that the TDNR model performed well by combining trust information with improved RMSE values compared to TDNR, which has no trust information.

## Comparison of TDNR with trust-based recommendation models

We have compared the TDNR framework with the following trust-based recommendation models:

**Trust-aware Recommender Systems (TARS):** TARS (*Bedi & Sharma, 2012*) is built on ACO, which creates an implicit trust graph using rating values. However, it does not consider explicit trust relationships when generating recommendations, which can negatively impact its performance in scenarios with sparse data. However, the TDNR model enhances the traditional trust-aware model by incorporating deep neural networks, allowing it to model complex nonlinear relationships between users and items. This results in improved recommendation accuracy, particularly in cold-start scenarios. TARS model is generally less computationally intensive than TDNR, making it more suitable for environments with limited computational resources.

**SoReg:** It incorporates social regularization into matrix factorization models, utilizing social trust networks to improve recommendation accuracy. TDNR's deep learning architecture allows it to capture more complex patterns in user interactions than the linear models used in SoReg. The inclusion of both trust and structural information further enhances its performance in informal e-learning environments. TDNR's deep learning approach requires more computational resources and may have longer training times than SoReg (*Ma et al., 2011*).

**Table 3 Comparison of TDNR with approaches address cold start problem.**

| Approach used | Cold start solution | Key features | Advantages | Limitations |
|---|---|---|---|---|
| Proposed TDNR | Combines deep neural networks with trust and structural information to predict recommendations for new users in e-learning environments. | – Tag modeling using LDA<br>– Trust degree computation between questioner and responder<br>– Structural information integration | – Effectively handles complex nonlinear relationships.<br>– Incorporates trust and structural information for enhanced accuracy.<br>– Excellent in cold-start scenarios. | – High computational complexity.<br>– Requires significant computational resources. |
| BasicMF (*Koren, 2008*) | Predicts ratings using matrix factorization based on user and item features. | – Decomposes user-item interaction matrix.<br>– Captures latent factors influencing preferences. | – Simple and effective.<br>– Widely used and easy to implement. | – Limited by linear assumptions.<br>– Doesn't handle cold-start or temporal dynamics well. |
| SVD (*Koren, Bell & Volinsky, 2009*) | Uses a temporal-based model combining neighborhood information and matrix factorization for cold start. | – Incorporates temporal dynamics.<br>– Uses neighborhood models. | – Adapts to changes in user preferences over time.<br>– More accurate than basic MF. | – Higher computational complexity.<br>– Requires more data for effective modeling. |
| SocialMF (*Jamali & Ester, 2010*) | Integrates social trust into matrix factorization to improve recommendation accuracy for new users. | – Uses social trust networks.<br>– Adjusts predictions based on trusted users. | – Effective in cold-start scenarios.<br>– Leverages trust for better recommendations. | – Relies on the availability of trust data.<br>– May struggle if trust data is sparse. |
| SoReg (*Ma et al., 2011*) | Uses social regularization in matrix factorization, enforcing similar preferences among socially connected users. | – Incorporates social relationships.<br>– Regularization to enforce similarity. | – Enhances accuracy by leveraging social context.<br>– Simple and effective for social networks. | – Assumes uniform influence across connections.<br>– Less effective in complex scenarios. |
| NNMF (*Dziugaite & Roy, 2015*) | Combines matrix factorization with neural networks to handle nonlinear interactions in cold start situations. | – Uses deep learning to model complex patterns.<br>– Captures nonlinear relationships. | – High predictive accuracy.<br>– Ideal for complex recommendation tasks. | – Computationally expensive.<br>– Risk of overfitting. |
| TrustCTR (*Ahmed et al., 2020*) | Integrates trust-aware recommendations with collaborative topic regression to handle cold starts. | – Combines topic modeling with trust.<br>– Uses both explicit and implicit trust. | – Contextually relevant recommendations.<br>– Effective in content-rich environments. | – Relies heavily on quality of topic modeling.<br>– Less effective with sparse content. |
| RSTE (*Ma, King & Lyu, 2009*) | Combines social trust and matrix factorization to create ensemble predictions for new users. | – Trust propagation.<br>– Hybrid of trust-based and collaborative filtering. | – Robust in social networks.<br>– Enhances accuracy in cold-start scenarios. | – Computationally intensive.<br>– Depends on social network data quality. |
| LOCABAL (*Tang et al., 2013*) | Uses local and global social context to provide recommendations in cold-start scenarios. | – Combines local (direct) and global (indirect) social contexts.<br>– Leverages broader social influence. | – Comprehensive social influence analysis.<br>– Accurate and relevant recommendations. | – Increased model complexity.<br>– Requires more computational resources. |

**SocialMF:** SocialMF (*Jamali & Ester, 2010*) combines matrix factorization with social trust propagation, aiming to improve recommendation accuracy by considering both user-item interactions and social trust relationships. While SocialMF uses trust relationships to enhance matrix factorization, TDNR's use of deep neural networks allows it to model nonlinear relationships more effectively. SocialMF is generally faster and less resource-intensive, making it more feasible for real-time applications. TDNR's higher computational complexity may limit its use in environments requiring quick recommendations.

**SocialFD:** SocialFD (*Yu et al., 2017*) applies matrix factorization combined with denoising techniques to address noise in trust data, aiming to improve the robustness and accuracy of recommendations. TDNR's use of deep learning allows it to naturally handle noise and outliers in the data, potentially offering better accuracy than SocialFD denoising approach. TDNR also benefits from its ability to incorporate both questioner-responder and responder-responder structural information. The SocialFD approach might be more efficient in scenarios with significant noise in the trust data, where TDNR's complexity could lead to longer processing times.

**Trust-aware Denoising Autoencoder (TbDAE):** TbDAE (*Ahmed et al., 2022*) integrates trust information into a denoising autoencoder to improve recommendation accuracy by learning robust latent representations. TDNR's architecture, which integrates deep neural networks with trust and structural information, allows it to model a wider range of interactions than TbDAE's focus on latent representation. TDNR's ability to address cold-start problems is also a significant advantage. TbDAE's focus on denoising may offer better performance in scenarios where data quality is an important issue. Additionally, TbDAE might be less computationally demanding than TDNR, making it more suitable for applications with resource constraints.

## Comparison of TDNR with approaches address cold start problem

The proposed TDNR model stands out in its ability to effectively tackle the cold-start problem in informal e-learning environments by incorporating both deep learning and trust-based methods. Compared to other approaches, it offers enhanced accuracy through its comprehensive modeling of nonlinear relationships and trust structures, though it requires significant computational power. Table 3 compares the TDNR model with existing approaches that address the cold start problem.

## CONCLUSIONS AND DISCUSSION

Informal e-learning has gained popularity during the last few years and has gotten more attention due to the lockdowns caused by the COVID-19 outbreak. E-learning, often known as online learning, has become essential for learners to acquire knowledge in their professional and social lives. The most common problems recommender systems face are data sparsity and cold start in e-learning systems. If a new questioner (learner) comes, how a recommender system can recommend an expert or answer the new questioner is challenging. Generating trust-based recommendations in the e-learning community is getting more attention nowadays. This article solved a learner cold-start problem by

incorporating trust and structural information about preferences. We designed a TDNR to incorporate questioner-responder and responder-responder structures. We proposed a TDNR model that models the complex non-linear relationships of the questioner and responder and their structural information based on trust. TDNR predicted the vote for the responder based on experts' trust. We evaluated TDNR on the Stack Overflow data using MAE and RMSE metrics and found reasonably good, dependable results. We have also compared TDNR with baseline algorithms and found improved results. The results depict that recommendation accuracy does not correlate with the quality of user experience.

Several factors support the validity of the TDNR framework. First, it builds on established concepts in collaborative filtering and deep learning, combining these with trust-aware mechanisms to tackle the cold-start problem. The framework incorporates latent Dirichlet allocation for tag modeling, which helps categorize questions accurately based on content, thereby improving the initial tagging process. This ensures that new questions are classified correctly, avoiding the issue of incorrect tag assignments, which can lead to irrelevant recommendations. Moreover, using a hierarchical temporal neural network (TNN) enables the framework to capture both questioner-responder and responder-responder interactions, enhancing the model's ability to understand and predict user preferences accurately. The framework has been validated using metrics such as RMSE and MAE, demonstrating its effectiveness in various e-learning scenarios.

The advantages of the TDNR model are that the TDNR framework integrates trust intensity between questioners and responders, significantly improving recommendations' accuracy, especially in addressing the learner cold-start problem. The model can better capture complex nonlinear relationships by incorporating questioner-responder and responder-responder structural information, leading to more accurate and personalized recommendations. Another advantage is that TDNR used the LDA for tag modeling, which helps categorize questions accurately, even when users input incorrect tags. This ensures that questions are appropriately classified and matched with relevant experts, improving the quality of recommendations.

One of the limitations of proposed TDNR model is that, due to its deep neural network architecture and the inclusion of multiple modules such as trust computation and tag modeling, it may have a high computational complexity. This can lead to longer processing times and require significant computational resources, which might not be feasible in all practical applications.

We implemented TDNR framework using python with TensorFlow, keras, pandas and other machine-learning libraries.

## Practical implications

### Enhanced learning experience in e-learning platforms

The TDNR model can significantly improve the user experience on e-learning platforms such as Coursera, Khan Academy, and Stack Overflow by providing personalized recommendations based on trust and deep learning mechanisms. For instance, in scenarios

where new users have little to no interaction history (a cold start), the TDNR model can still provide accurate course or content recommendations by leveraging the trust relationships within the platform. This helps new users find relevant resources more quickly, enhancing their learning journey.

### Application in community-driven platforms

In community-driven platforms like Stack Overflow, where users often seek help from experts, the TDNR model can identify and recommend the most trustworthy and relevant experts to answer new questions. By analyzing both the content of the questions and the trust relationships between users, the model can rank experts effectively, ensuring that questions are answered by those with the highest authority and reliability, thereby improving the quality of the responses and overall user satisfaction.

### Corporate training and knowledge management

Organizations that use internal e-learning systems for employee training and development can implement the TDNR model to tailor training materials to individual employees. By incorporating trust metrics, the model can recommend content that aligns with the employees' learning preferences and the expertise of their peers or mentors within the organization. This can lead to more effective and engaging training programs, ultimately enhancing skill development and productivity.

### A case study

Consider a large multinational corporation using an internal e-learning platform to train its employees. New employees, especially those in specialized roles, often struggle to find the most relevant training materials due to the vast content available. Implementing the TDNR model, the platform could analyze the trust relationships between new employees and their assigned mentors or colleagues. It would then recommend training modules that those trusted individuals have highly rated. As a result, new employees would receive more relevant and effective training recommendations, reducing the onboarding time and increasing productivity.

## Future work

In the era of information overload, there is a crucial need for recommendation systems to help learners discover better knowledge in informal e-learning environments.

In the future, we would like to incorporate questioner/responder context information to improve the recommendation quality. Further, the dynamics of the learners' communities can be investigated. One potential area for future research is integrating contextual information into the TDNR model. Currently, the model primarily relies on static trust relationships and user interactions without considering the broader context in which these interactions occur. By incorporating factors such as the interaction time and the specific learning environment, the model could provide even more personalized and relevant recommendations. Additionally, exploring dynamic trust relationships that evolve could significantly enhance the model's adaptability, allowing it to reflect changes in user behavior and social connections.

The limitation of proposed TDNR model is the high computational complexity associated with its deep learning architecture and the integration of multiple modules, which can lead to longer processing times and require substantial computational resources. Future research should aim to optimize the model's structure or develop more efficient algorithms to reduce this computational burden while maintaining or enhancing its accuracy. Another limitation is the model's reliance on the availability and accuracy of trust data, which may not be consistently available in all e-learning environments. This dependency could limit the model's applicability, so future research could explore alternative methods for handling trust data or develop techniques to infer trust relationships in environments where explicit data is scarce. Additionally, the current study's evaluation is primarily based on the Stack Overflow dataset, which, while representative, may not encompass the diversity of potential e-learning environments. Future studies should test the model across various datasets from different domains to ensure its generalizability.

### Funding
The funding of research is provided by Princess Nourah bint Abdulrahman University Researchers Supporting Project number (PNURSP2024R513), Princess Nourah bint Abdulrahman University, Riyadh, Saudi Arabia. AIDA Lab CCIS Prince Sultan University, Riyadh Saudi Arabia supported the Article Processing Charges (APC). The funders played a role in the study's design, providing essential infrastructure and computing equipment for the extensive experimental study. Additionally, we collaborated with the funders periodically to develop the methodology and fine-tune the parameters for the experimental outcomes. The funders had no role in the decision to publish, or preparation of the manuscript.

### Grant Disclosures
The following grant information was disclosed by the authors:
Princess Nourah bint Abdulrahman University Researchers Supporting Project: PNURSP2024R513.
Princess Nourah bint Abdulrahman University, Riyadh, Saudi Arabia.
AIDA Lab CCIS Prince Sultan University.
Riyadh Saudi Arabia supported the Article Processing Charges (APC).

### Competing Interests
The authors declare that they have no competing interests.

### Author Contributions
- Amjad Rehman conceived and designed the experiments, performed the experiments, analyzed the data, performed the computation work, prepared figures and/or tables, and approved the final draft.

- Adeel Ahmed conceived and designed the experiments, performed the experiments, analyzed the data, performed the computation work, prepared figures and/or tables, and approved the final draft.
- Tahani Jaser Alahmadi conceived and designed the experiments, analyzed the data, prepared figures and/or tables, authored or reviewed drafts of the article, and approved the final draft.
- Abeer Rashad Mirdad conceived and designed the experiments, prepared figures and/or tables, authored or reviewed drafts of the article, and approved the final draft.
- Bayan Al Ghofaily conceived and designed the experiments, prepared figures and/or tables, authored or reviewed drafts of the article, and approved the final draft.
- Khalid Saleem conceived and designed the experiments, performed the experiments, performed the computation work, prepared figures and/or tables, and approved the final draft.

## Data Availability

The code and dataset are available at GitHub and Zenodo:

- https://github.com/aahmedqau/TNN
- Adeel, A. (2024). Stack Overflow [Data set]. Zenodo. https://doi.org/10.5281/zenodo.13690863.

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
