# Peer review of "A framework for generating recommendations based on trust in an informal e-learning environment"

_PeerJ Computer Science, doi:10.7717/peerj-cs.2386_

## Round 0.1 · original submission · Major Revisions

Dear authors,

Thank you for submitting your article. Based on reviews' comments, your article has not yet been recommended for publication in its current form. However, we encourage you to address the concerns and criticisms of the reviewer and to resubmit your article once you have updated it accordingly. Before submitting the paper following should also be addressed:

1. References cited in text must appear in the reference list and vice versa. For example, Reference 64 does not exist in the References section.
2. Some paragraphs are too long to read. Long paragraphs should be divided into two or more for readability and comprehensibility.
3. In general, the literature review is not sufficient. More recent literature should be explored in depth. It is more of the type “researcher X did Y” rather than an authoritative synthesis assessing the current state-of-the-art. Advantages and disadvantages of the related works should be evaluated.
4. The paper lacks the running environment, including software and hardware. The analysis and configurations of experiments should be presented in detail for reproducibility. It is convenient for other researchers to redo your experiments and this makes your work easy acceptance. A table with parameter setting for experimental results and analysis should be included in order to clearly describe them.
5. Pros and cons of the methods should be clarified. What are the limitation(s) methodology(ies) adopted in this work? Please indicate practical advantages, and discuss research limitations.
6. Equations should be used with correct equation number. Please do not use “as follows”, “given as”, etc. Explanation of the equations should also be checked. All variables should be written in italic as in the equations. Their definitions and boundaries should be defined. Necessary references should be provided.
7. Some more recommendations and conclusions should be discussed about the paper considering the experimental results. The conclusion section is weak. There is also no discussion section about the results. It should briefly describe the results of the study and some more directions for further research. The authors should describe the academic implications, main findings, shortcomings and directions for future research in the conclusion section. The conclusion in its current form is generally confused. What will be happen next? What we supposed to expect from the future papers? So rewrite it and consider the following comments:
- Highlight your analysis and reflect only the important points for the whole paper.
- Mention the benefits
- Mention the implication in the last of this section.

Best wishes,

Reviewer 1 ·

Basic reporting

This manuscript ID-101479 presents a novel framework, Trust-aware Deep Neural Recommendation (TDNR), tailored to address the learner cold start problem in informal e-learning environments. The proliferation of information and data sharing in such environments poses challenges for recommending relevant answers to novice learners. TDNR innovatively models complex nonlinear relationships between learners and responses, leveraging latent Dirichlet allocation for tag modeling and utilizing hub-authority scores to rank experts relevant to specific tags. The framework further enhances recommendation accuracy by incorporating trust intensity between questioners and responders, as well as a relational graph derived from structural preferences. Evaluated on the Stack Overflow dataset, TDNR demonstrates significant improvements over rating-based and social-trust-based methods, showcasing its potential to facilitate the formation of vibrant informal e-learning communities. It was a pleasure reviewing this work and I can recommend it for publication in PeerJ Computer Science after a major revision. I respectfully refer the authors to my comments below.

Experimental design

1. The English needs to be revised throughout. The authors should pay attention to the spelling and grammar throughout this work. I would only respectfully recommend that the authors perform this revision or seek the help of someone who can aid the authors.
2. (References) Please adjust the style of all the references to meet the PeerJ Computer Science requirement.
3. (Page 19) The original figure 1 is not clear. Please redraw this figure clearly. Add some word in this figure, and indicate the stage and modules.

Validity of the findings

4. (In Section 1) The original sentence is suggested to revise as “the recommender systems related to education can provide personalized guidance to the learners [*], …”. ([1] "edmf: efficient deep matrix factorization with review feature learning for industrial recommender system," ieee tii, 2022; [2] "multi-perspective social recommendation method with graph representation learning," neurocomputing, 2022. [3] "carm: confidence-aware recommender model via review representation learning and historical rating behavior in the online platforms," neurocomputing, 2021.)
5. (Section 1 Introduction) The reviewer hopes the introduction section in this paper can introduce more studies in recent years. The reviewer suggests authors don't list a lot of related tasks directly. It is better to select some representative and related literature or models to introduce with certain logic. For example, the latter model is an improvement on one aspect of the former model.
6. (Figure 7-9) Experimental pictures or tables should be described and the results should be analyzed in the picture description so that readers can clearly know the meaning without looking at the body.

Additional comments

7. (In Section 2.2) The original sentence is suggested to revise as “deep learning has recently succeeded in collaborative filtering-based recommender systems [*], …” ([1] "deep variational matrix factorization with knowledge embedding for recommendation system," ieee tkde, 2021. [2] "deep matrix factorization with implicit feedback embedding for recommendation system," ieee tkde, 2019. [3] "a content-based recommendation algorithm for learning resources," multimedia systems, 2018.)
8. (Table I-II) All the values in this table should be with same data accuracy. The number of data after the decimal point are the same. Please check other Tables and section.
9. The authors are suggested to add some experiments with the methods proposed in other literatures, then compare these results with yours, rather than just comparing the methods proposed by yourself on different models.
10. Discuss the pros and cons of the proposed Recommendation algorithm models.

My overall impression of this manuscript is that it is in general well-organized. The work seems interesting and the technical contributions are solid. I would like to check the revised manuscript again.

Reviewer 2 ·

Basic reporting

The manuscript presents a thorough analysis of the TDNR model for trust-based recommendations in an informal learning environment using data from Stack Overflow. Here are the improvements needed for the following sections:
I recommend a major revision of this article to incorporate these suggestions and enhance the clarity and impact of the manuscript.

Experimental design

Materials and Methods
• Rationale for the Proposed Framework: Explain more clearly why deep neural networks and trust-sensitive systems are particularly suited for addressing the cold start problem.
• Tag Modeling Section: Ensure that the explanation of LDA and its application is not overly technical for readers unfamiliar with this method.
Clarify the Time Complexity Analysis
• Detail the Complexity: Expand on the time complexity analysis of Algorithm 1, especially for lines 9-16 where the complexity is mentioned as O(k×n×cost_per_training). Clarify what k and cost_per_training represent, and provide a more detailed breakdown if possible.
• Provide Examples: Include a specific example or case study to illustrate the complexity analysis in a practical scenario.

Validity of the findings

• Types of Trust: Discuss in more detail the types of trust information used in the TDNR model. Explain how trust is quantified and how it influences the recommendation process.
• Comparison with Other Trust Models: Compare the trust-based approach used in TDNR with other similar models. Highlight the advantages and limitations of your approach.
Cold Start Problem Solution
• Cold Start Approach: Provide a more comprehensive explanation of how the TDNR model specifically addresses the cold start problem. Include examples or case studies demonstrating the effectiveness of the approach.
• Comparison with Other Methods: Compare your cold start solution with those used in other models or approaches. Highlight the innovative contributions or improvements offered by your model.
Parameter Settings
• Details on Parameter Choices: Expand on why specific parameter values (e.g., learning rates, regularization rates) were chosen. Discuss the experiments that guided these choices.
• Parameter Tuning Insights: Include insights into how different parameter settings affect the model’s performance and possibly provide a range of recommended values.

Additional comments

Abstract
• Clarity and Conciseness: The abstract is generally clear. However, it would be beneficial to more explicitly highlight the significance of the results. For example, specify how the improvement in TDNR performance translates into practical benefits for users or systems.
Introduction
• Explanation of Learning Environments: The terms "formal e-learning environment" and "informal learning environment" are not explained clearly and concisely. A better definition of these concepts would help clarify the context for readers.
• Terminology Consistency: Ensure consistent use of terms such as "e-learning" and "informal learning." Currently, these terms are used interchangeably at times, which can create confusion.
• Supporting References: Some claims, such as those regarding the impact of information overload, could benefit from more recent and diverse references to underscore their relevance.

Related Work
• Relevance and Updates: Ensure that all references are up-to-date. Some references, particularly those from 2010, may be outdated. Include more recent studies to reflect current advancements in the field.
• Organization: Break down sections into smaller paragraphs to improve the readability of the content.
• Include a more thorough literature review to contextualize the TDNR model within the existing body of research. Discuss how your work builds on or diverges from previous studies.

Practical Applications and Future Work
• Real-World Applications: Discuss the practical implications of your results. How can the TDNR model be applied in real-world scenarios? Provide examples or case studies where the model could be beneficial.
• Future Work: Describe potential areas for future research or improvements. Discuss any limitations of your current study and how they could be addressed in future work.
• Quality of Figures: Ensure that figures are of high quality and clear enough to be understood.
• References: While the format of references is generally correct, some references are outdated. Use more recent references to improve the relevance of the bibliography.

I recommend a major revision of this article to incorporate these suggestions and enhance the clarity and impact of the manuscript.

---

## Round 0.2 · accepted · Accept

Dear authors,

Thank you for revising the manuscript. One reviewer did not respond to the invitation for reviewing the revision. Other reviewer thinks that his/her comments are well addressed. When I examined your revised article, it is thought that the necessary additions and arrangements have been performed according to the editor and reviewers' opinions and the article has been sufficiently improved. As such, the article is considered acceptable.

Best wishes,

Reviewer 1 ·

Basic reporting

The revised manuscript is improved compared to the former version. My previous comments are well addressed, and the presentation is improved significantly. The composition pattern and some other ideas are well elaborated, making them clearer. Overall, I tend to accept this manuscript.

Experimental design

The revised manuscript is improved compared to the former version. My previous comments are well addressed, and the presentation is improved significantly. The composition pattern and some other ideas are well elaborated, making them clearer. Overall, I tend to accept this manuscript.

Validity of the findings

The revised manuscript is improved compared to the former version. My previous comments are well addressed, and the presentation is improved significantly. The composition pattern and some other ideas are well elaborated, making them clearer. Overall, I tend to accept this manuscript.

Additional comments

The revised manuscript is improved compared to the former version. My previous comments are well addressed, and the presentation is improved significantly. The composition pattern and some other ideas are well elaborated, making them clearer. Overall, I tend to accept this manuscript.